# The Impact of Abusive Supervision on Job Insecurity: A Moderated Mediation Model

**DOI:** 10.3390/ijerph17217773

**Published:** 2020-10-23

**Authors:** Dawei Wang, Chaoyue Zhao, Yalin Chen, Phil Maguire, Yixin Hu

**Affiliations:** 1School of Psychology, Shandong Normal University, 88 Wenhua E Rd, Lixia District, Jinan 250014, China; wangdw@sdnu.edu.cn (D.W.); 2018020189@stu.sdnu.edu.cn (Y.C.); 2School of Psychology and Cognitive Science, East China Normal University, 3663 North Zhongshan Road, Shanghai 200062, China; 52203200019@stu.ecnu.edu.cn; 3Department of Computer Science, National University of Ireland, University Rd, H91 TK33 Galway, Ireland; pmaguire@cs.nuim.ie

**Keywords:** abusive supervision, job insecurity, leader-member exchange, power distance, social cognitive theory

## Abstract

This paper explores the impact of abusive supervision on job insecurity under the frameworks of the social cognitive theory and the leader-member exchange theory; additionally, it explores the mediating role of leader-member exchange (LMX) and the moderating role of power distance. In this study, 944 employees from two state-owned enterprises located in China were surveyed via questionnaires. Results of the correlation analysis and statistical bootstrapping showed that (i) abusive supervision was significantly and positively related to job insecurity, (ii) LMX played a mediating role in the impact of abusive supervision on job insecurity, and (iii) power distance played a moderating role in the relationship between LMX and job insecurity. Based on the social cognitive theory, this study broadens the perspective of studies regarding job insecurity. It also provides practical suggestions for avoiding abusive supervision and for alleviating employees’ insecurities about management.

## 1. Introduction

The past few decades have been characterized by continual changes in people’s working conditions. When confronted with a competitive economic environment, large numbers of organizations have implemented downsizing, restructuring, reorganization, or relocation policies to enhance organizational effectiveness and reduce expenditures [1]. In addition, the development of various high-tech startups requires employees to have greater knowledge of technology, which provides new challenges when employees are completing their work [2]. Given this situation, employees feel anxious due to the increasing instability in their line of work; they are fearful about their career prospects and experience a strong sense of job insecurity. Nowadays, the widely accepted definition of job insecurity is “uncertainty regarding stable and continuous employment.” It should be noted that job insecurity is only a subjective perception and evaluation of the risk of unemployment; however, it does not mean that employees are actually in danger of losing their jobs [3]. Existing studies have shown that job insecurity significantly affects employees’ psychological and physical health over time [4]. Because job insecurity is an important source of stress, its mechanism and mitigation strategies deserve research attention. Previous studies have focused more on the consequences and harm caused by job insecurity, whereas few have explored its antecedent variables [5]. Based on social cognitive theory, the aims of our study were to investigate the causes of job insecurity and to probe into its specific mechanisms.

According to social cognitive theory, the employees’ cognitive processing can be affected by situational variables, which in turn produce different emotional responses [6]. As a subjective feeling, job insecurity is inevitably affected by the external environment. One example—leadership, a distinctly important factor in the work environment—is certain to have an impact on employees’ insecurities. Previous studies have focused more on positive leadership, such as ethical leadership, which has been shown to play an important role in reducing employees’ sense of job insecurity [7]. At the same time, authentic leadership can reduce job insecurity by increasing psychological empowerment and psychological capital [8]. However, not all leadership behaviors are positive. As an example of negative leadership, abusive supervision has attracted the attention of researchers [9,10], and has been shown to be associated with many negative psychological outcomes [11]. Accordingly, based on social cognitive theory, we hypothesize that abusive supervision impacts employees’ job insecurities.

In addition to verifying the relationship between abusive supervision and job insecurity, this study attempts to explore its underlying mechanism. Abusive supervision, in the form of hostile behavior from managers, tends to make employees lose their trust, loyalty, and respect for their leaders. This also confines employees’ social exchanges with leaders to taking place purely within economic or contractual exchanges. The lower the quality of their social exchanges, the more stressful employees will become; furthermore, they will experience a stronger sense of job insecurity [12]. Above all, we argue that leader-member exchange (LMX) may play a mediating role in the relationship between abusive supervision and job insecurity.

If abusive supervision promotes job insecurity through LMX, the question arises about whether this effect is influenced by other individual factors. Previous studies have shown that cultural value orientation influences an individual’s assessment of an event and, thus, the individual’s subjective perceptions [13,14]. Power distance is one of the most important cultural values that can be found in almost all existing cultural value frameworks [15]. It reflects the degree of individual acceptance of the imbalance of power distribution among organizations, institutions, and societies [16,17,18]. Employees with different levels of power distance may have different interpretations of the same relationships with leaders; that is, employees with high power distance may obey authority, while employees with low power distances will pursue a balance in status, thus affecting their sense of job insecurity [16]. Accordingly, we propose that power distance can moderate the relationship between LMX and job insecurity.

## 2. Theory and Hypotheses

### 2.1. Abusive Supervision and Job Insecurity

Though development of technology and globalization of international markets have brought many benefits to society and enterprises, such advances place greater requirements on employees; as a result, more of them are experiencing a sense of job insecurity [19]. Job insecurity refers to employees’ perceptions and interpretations of the work environment, their expectations about risks regarding the continuity of work, and anxiety and uncertainty over their existing jobs [3]. Job insecurity leads to negative attitudes toward work and the organization as well as reduced job satisfaction, job involvement, job performance, and organizational commitment; it also causes irreversible damage to employees’ physical and mental health [3,20]. In the Chinese context, job insecurity also brings many harms, such as emotional exhaustion and increased employee turnover intention [21]. A series of negative consequences resulting from job insecurity have led an increasing number of scholars to focus on its antecedents and mitigation strategies. Through a meta-analysis, Shoss (2017) pointed out that leadership type, as a key organizational factor, could not be ignored given its impact on job insecurity.

Abusive supervision, as a typical negative leadership behavior, has attracted many scholars to explore its antecedents and consequences. According to Tepper (2000), abusive supervision is defined as “subordinates’ perceptions of the extent to which supervisors engage in the sustained display of hostile verbal and nonverbal behaviors, excluding physical contact.” Examples include abusing subordinates in front of others, withholding important information, and threatening and intimidating subordinates [22]. Abusive supervision can lead to a series of negative consequences, such as reduced job satisfaction, decreased job performance, and increased turnover [5]. In addition, employees working under conditions involving long-term abusive supervision are prone to suffering increased stress, which leads to emotional exhaustion, job burnout, anxiety, and other negative psychological outcomes [11].

The relationship between abusive supervision and job insecurity can be explained by social cognitive theory. The basic assumption of this theory is that there is continuous interaction between the external environment, cognitive factors, and human behavior, which varies as those factors change [23]. When this relationship is unbalanced or uncoordinated, the individual feels psychological pressure [24]. To release such pressure, employees restore balance and coordination by implementing external behaviors or by adjusting their internal cognitions [25]. Specifically, when exposed to pressure from managers’ abusive supervision, employees will reconstruct their self-cognition and behaviors in order to adapt to the situation. They may lose their sense of belonging and perceive betrayal and distrust from their leaders, which will lead to concerns about work continuity and job insecurity. Given these arguments, we propose the following:
**Hypothesis** **1.**Abusive supervision is positively related to job insecurity.

### 2.2. The Mediating Role of Leader-Member Exchange (LMX) 

Graen and Dansereau first proposed LMX theory in 1972. Later, Liden et al. (1998) proposed that the theory should include exchanges involving work and society [26,27]. Since then, social exchange theory has been introduced into interpretations of LMX and the relationship between leaders and employees is regarded as a kind of social exchange [28]. Liden considered that LMX theory is usually expressed in two distinct connotations. One is an economic or contractual exchange between leaders and employees; it does not exceed the requirements of employment contracts. When abusive supervision occurs, mutual trust, loyalty, and responsibility cannot be established between leaders and employees. Employees become “outsiders” who are in low-quality LMX relationships with their leaders. Abusive supervisory behavior includes ridicule and neglect of employees, which may cause employees to perceive that their expectations of affective exchanges have been violated [29]. The resulting poor LMX relationships between employees and leaders are only limited to the scope of formal work [26,30]. The other connotation refers to social exchanges between leaders and their employees that are beyond the formal contract and based on mutual trust, loyalty, and responsibility [31,32]. Based on this type of relationship, employees who accomplish their work tasks and perform well expect to receive positive feedback and support for their work [33]. We believe that leader’s abusive supervision, as a hostile behavior towards employees, undermines subordinates’ expectations regarding trust, loyalty, and mutual responsibility, meaning that leaders and subordinates form poor social exchange relationships.

According to LMX theory [12,34], a one-on-one relationship between each employee and leader is established through repeated daily interactions. However, due to the different daily interactions between leaders and employees, each dyadic relationship is different, resulting in differentiated exchange relationships [12,26]. In-group members can acquire mutual respect, deepening trust, and the development of a business relationship in social exchanges with the leader [35]. At the same time, in-group members get more attention and support from leaders and enjoy more resource allocation [36]. In addition, their jobs are more stable and they get more opportunities for promotion [37]. Conversely, subordinates in the out-group face a different situation [37,38]. Low-quality LMX limits their performance to that specified in formal job descriptions; moreover, they often do not have access to extra resources in their work, do not get the support of supervisors, and do not enjoy the same promotion opportunities, leading them to feel more negatively about their jobs [37,39]. Low-quality LMX limits their performance to that specified in formal job descriptions; moreover, they tend to be given ordinary assignments, to obtain less supervisory support, to feel more negatively about their jobs, and to have fewer opportunities for promotion [37,39]. Erdogan and Enders (2007) show that improvement of LMX can boost employees′ job satisfaction and work performance and that this relationship is moderated by supervisors′ perceived organizational support [40]. At the same time, Yildiz (2018) suggested that LMX reduces the turnover intention of employees by reducing mobbing behaviors [41]. In addition, Xu et al. (2015) suggested that LMX can be used as a moderator variable to mitigate the negative effects of improper supervision [42]. In sum, compared with in-group subordinates, out-group subordinates with low-quality LMX are more likely to experience job insecurity. In light of these findings, we propose the following hypothesis:
**Hypothesis** **2.**Leader-member exchange (LMX) mediates the relationship between abusive supervision and job insecurity.

### 2.3. The Moderating Role of Power Distance

Based on the analysis described above, we speculated that abusive supervision influences job insecurity among employees through LMX, but that this mechanism may be affected by individual factors. In the social cognitive mechanism, the extent to which individuals experience strain resulting from stressors depends on how they interpret those stressors [13]. In other words, even in the same stressful situation, different interpretations of stressors can produce different results. 

At the individual level, power distance refers to the extent to which one accepts the legitimacy of unequally distributed power in institutions, organizations, and societies. Consequently, individuals oriented toward high power distance are more likely to accept an imbalance in the distribution of power. On the contrary, individuals with low power distance cannot accept an imbalance in the distribution of power [16,43,44]. In the initial research on cultural values, scholars generally regarded it as a social variable [44]; subsequent research, however, has shown that each of its value dimensions vary widely among individuals in different societies [16,45], and that these variations have a significant impact on job insecurity [5].

Consequently, we propose that employees with different levels of power distance can have dissimilar outcomes—even if the quality of relationships with their leaders is the same—because of their different acceptance levels regarding an imbalance in power distribution [46]. Specifically, individuals with high power distance recognize the existence of a hierarchy and show deference and obedience to those in authority [47]. They regard themselves as inferior to their leaders in terms of organizational status, and take for granted the low quality of their relationships with their leaders [48]. By taking low-quality LMX for granted, they do not have a great deal of psychological stress and do not perceive that their jobs are threatened, so they will typically not have a sense of job insecurity. 

In contrast, individuals with low power distance seek equal status within their organization. Having a low-quality relationship between leaders and such employees aggravates the unequal relationship between the two parties and conflicts considerably with employees’ beliefs. Therefore, employees with low power distance will evaluate a low-quality relationship as a high-level threat, which will seriously affect the stability and sustainability of their work [7]. Based on these arguments, we propose the following hypotheses:
**Hypothesis** **3.**Individual power distance moderates the relationship between abusive supervision and job insecurity, such that the relationship will be weaker for employees who have higher levels of power distance.

Hypotheses 2 and 3 represent a moderated mediating effect—that is, LMX mediates the relationship between abusive supervision and job insecurity—but the magnitude of this mediating effect is affected by the power distance of employees. Specifically, the indirect relationship between abusive supervision and job insecurity is weaker for employees who have higher degrees of power distance. Accordingly, we propose the following:
**Hypothesis** **4.**An individual’s power distance moderates the relationship between abusive supervision and job insecurity via LMX, such that the indirect effect is weaker for employees who have higher levels of power distance.

## 3. Method

### 3.1. Sample and Procedures

Participants in this study were 936 front-line employees from two state-owned enterprises located in China. The study was conducted in accordance with the Declaration of Helsinki, and the protocol was approved by the Ethics Committee of Shandong Normal University. The ethical approval project identification code is SDNU2020006. Informed consent was obtained from leaders and employees of each company. Information pertaining to all participants was kept strictly confidential, with each participant reserving the right to withdraw from the study at any time. 

In total, 1050 questionnaires were distributed and 936 were returned (89.1% response rate). Among these participants, 64.7% were female and 35.3% were male; 83.5% were married, 14.5% were unmarried, and 1.9% reported their status as “other.” In terms of age, 6.8% were under 25 years old, 32.9% were 26–30 years old, 22.6% were aged 31–35 years, 9.4% were aged 36–40, 25.7% were aged 41–50, and 2.5% were over the age of 50. In terms of academic qualifications, 1% had completed junior high school, 35.6% had a high school diploma, 42.0% had an associate degree, 20.3% had a bachelor’s degree, and 1.2% had a master’s degree. In terms of work experience, 0.6% of the participants had worked for less than 1 year, 5.9% worked for 1–3 years, 21.7% worked for 4–6 years, 20.3% worked for 7–9 years, and 51.5% worked for more than 10 years.

### 3.2. Measures

#### 3.2.1. Abusive Supervision

The abusive supervision scale was developed by Tepper (2000) and revised by Huang (2012); it consists of 10 items and two dimensions (i.e., “ridicule” and “neglect concealment”) [49]. Ridicule includes six items, such as “My supervisor is rude to me.” Neglect concealment includes four items, such as “Although I work hard, my supervisor will not praise me.” Responses were measured according to a five-point Likert format ranging from 1 (*strongly disagree*) to 5 (*strongly agree*). The Cronbach’s alpha coefficient was 0.97.

#### 3.2.2. Leader-Member Exchange (LMX)

We adopted the LMX-7 scale developed by Janssen and Van Yperen (2004) [50]; it contains seven items, including “My working relationship with my supervisor is effective.” A five-point Likert format was used, with responses ranging from 1 (*strongly disagree*) to 5 (*strongly agree*). The Cronbach’s alpha coefficient was 0.95.

#### 3.2.3. Power Distance

We assessed power distance using a six-item measure developed by Dorfman and Howell (1988) [51]. An example item is “Managers should make most decisions without consulting subordinates.” A seven-point Likert format was used, with responses ranging from 1 (*strongly disagree*) to 7 (*strongly agree*). Higher scores represent a higher power distance orientation. The Cronbach’s alpha coefficient was 0.88.

#### 3.2.4. Job Insecurity

Job security was measured using the seven-item scale established by Borg and Elizur (1992) and revised by Staufenbie and König (2011). Its two dimensions are cognitive job insecurity and affective job insecurity [52,53]. Cognitive job insecurity includes four items, such as “My job is secure.” Affective job insecurity includes three items, such as “The thought of losing my job troubles me.” A seven-point Likert format was used, with responses ranging from 1 (*strongly disagree*) to 7 (*strongly agree*). The Cronbach’s alpha coefficient was 0.83.

#### 3.2.5. Control Variables

Previous studies have shown that employees′ demographic characteristics (such as gender, age, etc.) can have an impact on their job insecurity [4,7]. Hence, we included gender, marriage, age, education, and working years as control variables in our analysis.

## 4. Results

### 4.1. Common Method Bias

This study adopted the Harman single factor technique to estimate the influence of common method bias. The results showed that five factors emerged, with an interpretation rate of the population variance of 76.43%. The interpretation rate for the first common factor was 36.98%, indicating that there was no serious common method bias in this study [54,55]. 

### 4.2. Confirmatory Factor Analyses

We used Mplus 7.0 to conduct the confirmatory factor analyses (CFAs). The measurement model fitted the data acceptably (*χ^2^* (380) = 922.23, *p* < 0.001, Root Mean Square Error of Approximation (RMSEA) = 0.04, comparative fit index (CFI) = 0.98, Tucker-Lewis index (TLI) = 0.98, and standardized root mean square residual (SRMR) = 0.06). We also examined several alternative measurement models and compared them with the four-factor model. As shown in Table 1, the four-factor model fits our data better than other models, suggesting that the study respondents could distinguish the focal constructs clearly.

### 4.3. Correlation Analysis

Table 2 shows means, standard deviations, and correlations of the study variables. The results showed a significant and negative correlation between abusive supervision and LMX (*r* = −0.42, *p* < 0.01) and a significant and positive correlation between abusive supervision and job insecurity (*r* =
0.35, *p* < 0.01). Additionally, LMX had a significant and negative correlation with job insecurity (*r* =
−0.35, *p* < 0.01). Therefore, Hypothesis 1 is supported.

### 4.4. The mediating Effect of LMX

SPSS 25.0 and PROCESS 3.2 were used to test the mediating effect of the data. First, results of the regression analysis for job insecurity and abusive supervision indicated that abusive supervision had a significant and positive influence on job insecurity (*β* = 0.45, *p* < 0.001). Then, with LMX as the mediator and gender, marriage, age, working years, and education as control variables, the bootstrap test was performed 5000 times with a 95% confidence interval. The results in Figure 1 show that abusive supervision had a significant and negative influence on LMX (*β* = −0.41, *p* < 0.001) and a significant, positive influence on job insecurity (*β* = 0.30, *p* < 0.001). Furthermore, LMX had a significant and negative influence on job insecurity (*β* = −0.34, *p* < 0.001). At the same time, the 95% confidence interval for the indirect effect of abusive supervision on job insecurity through LMX was [0.09, 0.20]. These results indicate that LMX played a partial mediating role in the relationship between abusive supervision and job insecurity; thus, Hypothesis 2 is supported.

### 4.5. Moderated Mediating Effect

PROCESS 3.2 was used to perform the moderated mediating effect test on the data; Model 14 was selected, and the bootstrap test was conducted 5000 times, with a 95% confidence interval (CI). As shown in Table 3, the results showed that abusive supervision had a significant and negative impact on LMX (*β* = −0.41, *p* < 0.001) and that LMX had a significant and negative impact on job insecurity (*β* = −0.35, *p* < 0.001). The interaction between power distance and LMX had a significant impact on job insecurity (*β* = 0.13, *p* < 0.01), indicating that the moderated mediating effect was significant and, hence, that Hypothesis 3 is supported.

We also examined the conditional indirect effect of abusive supervision on job insecurity via LMX at varying power distance levels (one SD above the mean and one SD below the mean). The conditional indirect effect for abusive supervision via LMX on employees’ job insecurity was 0.10 with a 95% CI of [0.04, 0.15] for higher power distance, as opposed to 0.19 with a 95% CI of [0.11, 0.27] for lower power distance. The difference between these indirect effects for the two conditions was −0.09 with a 95% CI of [−0.18, −0.01]. These results exposed a significant moderating effect of power distance on the indirect effect, thus supporting Hypothesis 4.

We then employed procedures of Aiken and West (1991) to plot the pattern of significant interaction effects [56]. High power distance was designated as one SD above the mean, while low power distance was designated as one SD below the mean. Consistent with our expectation, as depicted in Figure 2, the negative relationship between LMX and job insecurity was relatively stronger for employees with lower power distance. This finding further validates Hypotheses 3 and 4.

## 5. Discussion

### 5.1. Theoretical Contributions

First, the findings of this study verified that abusive supervision significantly and positively affects job insecurity. Our result is currently one of the few validations of this relationship. Abusive supervision, as a negative leadership behavior, is produced when a leader ignores the feelings of employees and deals with them with excessive words or abusive behavior [9,42], thus creating stressful situations for employees and forcing them to adjust their cognition and behavior in order to achieve consistency with the environment [48]. Employees cannot take actions to resist because of the gap separating them from their supervisors, thus increasing the psychological pressure on them and stoking concerns about the continuity of their work and access to important work resources—ultimately, resulting in job insecurity [29]. Previous studies have tended to focus on the impact of positive leadership behavior on job insecurity, advocating an increase in such behavior to alleviate job insecurity [7,57]. This study, in contrast, has focused on negative leadership behavior—namely, abusive supervision—and has demonstrated a positive correlation between such leadership and job insecurity. This focus not only enriches the study of the antecedent variables of job insecurity but also provides a new perspective for future research.

Second, the results of this study indicate that LMX plays a mediating role in the relationship between abusive supervision and job insecurity. As well as clarifying this relationship, we have explored the internal mechanisms and psychological processes behind it. We took social cognitive theory as the main body, combined it with LMX theory, and confirmed that LMX is an effective transmission mechanism for the impact of abusive supervision on job insecurity. Specifically, employees subjected to abusive supervision may perceive a decline in LMX, thereby affecting job insecurity. LMX theory assumes that there are two different kinds of relationships between leaders and subordinates: high-quality and low-quality, or “insiders” and “outsiders” [58]. Abusive supervision, such as ridicule and neglect concealment, is a negative leadership behavior [9,59]; it will inevitably lead to a reduction in social exchanges between leaders and employees, allowing relationships to become even more alienated. Such low-quality LMX can make employees feel a certain degree of betrayal and distrust [60], so that they will subjectively define themselves as “outsiders,” leading to job insecurity. Although previous studies have explored the impact of LMX on turnover intention, the results of this study suggest that LMX is more likely to be a mediation mechanism that translates leaders’ negative behavior into employee’s job insecurity. By combining social cognitive theory with LMX theory [23,61], we have clarified the internal mechanism of abusive supervision that affects employees’ job insecurities.

Finally, the results of this study have shown that power distance moderates the effect of LMX on job insecurity. According to social cognitive theory, the impact of stressors on individuals depends on how those stressors are interpreted [62]. Previous studies have found that an individual’s cultural value orientation is an important factor affecting job insecurity [5]. Therefore, abusive supervision affects LMX and, subsequently, job insecurity, which inevitably has some boundary conditions. Through our investigation, we have found that power distance plays a negative moderating role in the relationship between LMX and job insecurity. That is, the higher the power distance of employees, the smaller the negative impact of LMX on job insecurity. This discovery not only expands the research on abusive supervision and job insecurity but also clarifies the boundary conditions whereby abusive supervision affects LMX and job insecurity. In addition, this study has also explored individual factors for job insecurity from the perspective of cultural values, an aspect which deserves further attention in future research.

### 5.2. Practical Significance

First, the results of this study show that abusive supervision can positively influence a sense of job insecurity among employees. This indicates that, in order to alleviate job insecurity, organizations should first consider avoiding abusive supervision. To this end, scientific systems and mature process management can be used to restrain managers’ abusive behavior and to avoid the abuse of power. For example, 360° feedback can give employees the power to evaluate their supervisors and to influence supervisors’ performance pay. At the same time, leadership training—such as setting up leadership training courses and role-playing, exploring which behaviors will be regarded as abusive supervision by subordinates and which behaviors can be used to improve morale—is also an effective strategy. In this way, managers can effectively adopt positive leadership behaviors in the future and reduce the incidence of abusive supervision within their organizations [63].

An interesting observation in our study was that men were more perceptive of abusive supervision than women, replicating the findings of previous studies [64]. Such greater sensitivity of men to abusive supervision may have roots in traditional Chinese concepts, which associate male self-esteem with workplace responsibility [65], thereby magnifying the impact of criticism or neglect by leaders. 

Second, in view of the mediating role of LMX in abusive supervision and job insecurity, managers must strengthen their relationships with subordinates. In addition to economic exchanges with employees, leaders should pay more attention to social exchanges. Moreover, organizations should be fair and impartial in establishing rules and regulations. Leaders should be open, honest, and respectful toward subordinates so that subordinates can fully understand relevant information about the organization. These measures can make employees feel that the organizational atmosphere is fair and equitable and that their relationships with leaders are satisfactory, effectively reducing job insecurity.

Finally, our results have shown that power distance moderates the relationship between LMX and job insecurity. That is, employees with lower power distance are more likely to experience a stronger sense of job insecurity if they work under the condition of abusive supervision, which is associated with a decline in LMX. Therefore, managers should create an equitable and harmonious organizational atmosphere, eliminate red tape, and maintain smooth communication channels between subordinates and leaders. At the same time, organizations should implement participatory management (i.e., more employees should be involved in management). Together, these measures can contribute to the formation of psychological security for employees, can promote the development of relationships between superiors and subordinates, and can reduce the job insecurity of employees.

## 6. Limitations and Future Directions

First, our data collection was carried out during a single time period; therefore, it might have been susceptible to common method bias [54]. Although the relevant empirical data support the theoretical hypotheses of this study, we recommend the use of multiple time points for data collection in future research in order to minimize the impact of common method bias effectively [54].

Second, because of industry constraints, most participants in this study were women. Moreover, all data were collected in state-owned enterprises in China, which limits the external validity of our findings. For instance, the prevailing managerial style in a society or country could be significantly influenced by the current situation in that country. Therefore, future research should expand the number and diversity of subjects and explore whether the relationships identified here are observed in other industries and in a cross-cultural context.

Third, according to social cognitive theory, different individuals have diverse perspectives and approaches for looking at problems. Because of different perspectives held by managers and employees, actions taken by managers to safeguard organizational interests or to strengthen employee management may be regarded as negative and hostile by some employees; such actions may also be regarded as examples of abusive supervision. Therefore, we recommend that future studies include multi-source data from both subordinates and their supervisors so that hypothesized relationships can be tested more rigorously.

In this study, we explored the impact of abusive supervision on job insecurity and analyzed the impact mechanism in considerable depth. However, we believe that the variables affecting job insecurity are not limited to LMX and power distance; there are likely many other factors that should be investigated. Future studies might explore further impact mechanisms or include cross-level analyses at the organizational level.

## 7. Conclusions

The results of our study have shown that abusive supervision has a positive impact on job insecurity and that LMX mediates the relationship between inappropriate supervision and job insecurity. In other words, abusive supervision as perceived by employees negatively affects LMX, thus increasing employees’ experience of job insecurity. In addition, power distance plays a moderating role in the relationship between LMX and job insecurity—that is, the lower the power distance of employees, the stronger the negative correlation between LMX and job insecurity.

## Figures and Tables

**Figure 1 ijerph-17-07773-f001:**
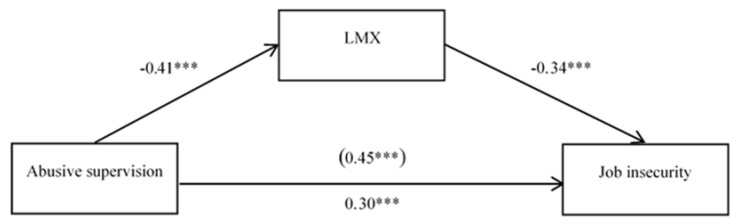
The mediating role of LMX in the relationship between abusive supervision and job insecurity. *** *p* < 0.001.

**Figure 2 ijerph-17-07773-f002:**
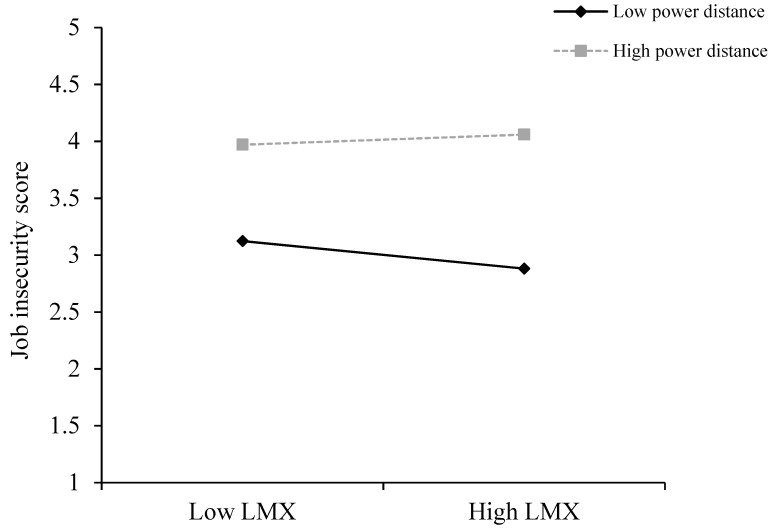
Interaction effect of LMX and power distance on job insecurity.

**Table 1 ijerph-17-07773-t001:** Results of confirmatory factor analysis of the measurement models.

Measurement Models	*χ^2^*	*df*	*χ^2^*/*df*	RMSEA	CFI	TLI	SRMR
Four-factor (A,B,C,D)	922.23	380	2.43	0.04	0.98	0.98	0.06
Three-factor (A,B+C,D)	3173.12	383	8.28	0.09	0.90	0.88	0.13
Two-factor (A+B+C,D)	5284.89	385	13.73	0.12	0.82	0.80	0.15
One-factor (A+B+C+D)	6632.72	386	17.18	0.13	0.77	0.74	0.16

Note: A = abusive supervision, B = leader-member exchange (LMX), C = power distance, and D = job insecurity.

**Table 2 ijerph-17-07773-t002:** Scale descriptive statistics and Pearson correlations among study variables (N = 936).

	M	SD	1	2	3	4	5	6	7	8	9
1 gender	1.35	0.48	-								
2 marriage	1.18	0.43	−0.13 **	-							
3 age	3.22	1.38	0.55 **	−0.26 **	-						
4 education	2.85	0.79	−0.16 **	−0.01	−0.14 **	-					
5 working years	4.16	1.00	0.45 **	−0.29 **	0.73 **	−0.04	-				
6 abusive supervision	1.77	0.87	0.30 **	−0.00	0.25 **	−0.10 **	0.22 **	-			
7 LMX	3.89	0.75	0.01	−0.01	−0.01	0.06	−0.03	−0.42 **	-		
8 power distance	2.30	0.85	0.15 **	−0.01	0.13 **	−0.05	0.13 **	0.42 **	−0.15 **	-	
9 job insecurity	3.40	1.10	0.07 *	0.05	0.11 **	−0.20 **	0.06	0.35 **	−0.35 **	0.20 **	-

Notes: SD = Standard deviation. N = 936. Gender was coded as “1” for women and “2” for men. Marriage was coded as “1” for married and “2” for unmarried. Education was coded as “1” for junior high school diploma, “2” for high school diploma, “3” for associate degree, “4” for undergraduate diploma, and “5” for master diploma. Age was coded as “1” for under 25 years old, “2” for 26–30 years old, “3” for 31–35 years old, “4” for 36–40 years old, “5” for 41–50 years old, and “6” for over 50 years old. Working years was coded as “1” for less than 1 years, “2” for 1–3 years, “3” for 4–6 years, “4” for 7–9 years, and “5” for more than 10 years. * *p* < 0.05; ** *p* < 0.01.

**Table 3 ijerph-17-07773-t003:** Moderated mediating effect.

	Outcome: LMX			Outcome: Job Insecurity		
	*β*	SE	*t*	*β*	SE	*t*
Gender	0.21	0.06	3.74 ***	−0.13	0.08	−1.61
Marriage	0.02	0.05	0.39	0.14	0.08	1.81
Age	0.04	0.03	1.40	0.09	0.04	2.33 *
Education	0.04	0.03	1.55	−0.22	0.04	−5.22 ***
Working years	−0.02	0.03	−0.69	−0.04	0.05	-0.91
Abusive supervision	−0.41	0.03	−15.09 ***	0.25	0.05	5.34 ***
LMX				−0.35	0.05	−7.24 ***
Power distance				0.09	0.04	2.09 *
LMX*Power distance				0.13	0.05	2.84 **
R^2^	0.20			0.21		
F	39.29 ***			28.11 ***		

Note: *** *p* < 0.001; ** *p* < 0.01; and * *p* < 0.05.

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
