# Peer review of "The Impact of Abusive Supervision on Job Insecurity: A Moderated Mediation Model"

_ijerph, 2020, doi:10.3390/ijerph17217773_

Round 1

Reviewer 1 Report

Comments to content of article:

Formal comments:

  1. There is inaccuracy with the resource “Hofstede, 1980. There are two articles in the list of references from the same year. It is not clear which is mentioned in the text. The solution is to describe these references as Hofstede, 1980a and Hofstede, 1980b.
  2. This resource Hofstede mentioned on line 159 is presented without the year of publishing.
  3. In table 2 the word “supervision” is incorrectly divided.
  4. SD abbreviation is not explained in the text neither in the table 2.
  5. The Measurement Models presented in the table 1 is divided incorrectly. It will be better to present four factors, three factors etc. on one line and (A,B,C,D) or (A,B+C,D) etc. on the second line in the table.
  6. Information in percent (like 64,7%) is usually presented separately (like 64,7 %) with the space between number and %.

Content comments:  

  1. Results of research present in table
  2. In the table 2 and 3 are presented analysis of variables (gender, age, education, …). It will be useful to include in description of results or in discussion the differences of the results based on these variables.

Authors focus on impact of abusive supervision on job insecurity; additionally, they analyse the role of leader-member exchange (LMX) and the moderating role of power distance. The literature review explains the problem sufficiently, the formulations of the hypotheses are clear. The results mainly confirm the outcomes of other previous analysis in this area, it brings confirmation of negative role of abusive supervision in management. There are mentioned limits of this research, I recommend to include in these limits or in future research the confirmation of the results in public/private sector and in different countries. The atmosphere in the society and in the companies could be influenced also by the situation in the country.

Author Response

Thank you for taking time to provide this in-depth review of our paper. This valuable information has allowed us to improve the manuscript. Below, we clarify the changes that have been made.

  1. There is inaccuracy with the resource “Hofstede, 1980. There are two articles in the list of references from the same year. It is not clear which is mentioned in the text. The solution is to describe these references as Hofstede, 1980a and Hofstede, 1980b.

Response: Thank you for your kind advice. We have made the recommended modification, the modified parts have been marked red. [p14, line512-514]

  1. Hofstede, G. (1980a). Culture's consequences: International differences in work-related values. Beverly Hills, CA: Sage.
  2. Hofstede, G. (1980b). Culture and organizations. International Studies of Management & Organization, 10(4), 15-41.

  1. This resource Hofstede mentioned on line 159 is presented without the year of publishing.

Response: We've changed it to (Hofstede, 1980b) [p4, line172].

  1. In table 2 the word “supervision” is incorrectly divided.

Response: We adjusted the format and marked it in red [p8, line275].

  1. SD abbreviation is not explained in the text neither in the table 2.

Response: SD = Standard deviation. Notes have been added to the article and changes have been marked in red [p8, line276].

  1. The Measurement Models presented in the table 1 is divided incorrectly. It will be better to present four factors, three factors etc. on one line and (A,B,C,D) or (A,B+C,D) etc. on the second line in the table.

Response: We have adjusted the format and marked it in red [p6, line267].

  1. Information in percent (like 64,7%) is usually presented separately (like 64,7 %) with the space between number and %.

Response: We have made the recommended modification, the modified parts has been marked red.

  1. In the table 2 and 3 are presented analysis of variables (gender, age, education, …). It will be useful to include in description of results or in discussion the differences of the results based on these variables.

Response: Thank you for your kind advice. We have rewritten this and clarified it as follows:

An interesting observation in our study was that men were more perceptive of abusive supervision than women, replicating the findings of previous studies (Wang, Zhou, Maguire, Zong, & Hu, 2019). Such greater sensitivity of men to abusive supervision may have roots in traditional Chinese concepts, which associate male self-esteem with workplace responsibility (Chen, 2003), thereby magnifying the impact of criticism or neglect by leaders. [p11, line385].

  1. There are mentioned limits of this research, I recommend to include in these limits or in future research the confirmation of the results in public/private sector and in different countries. The atmosphere in the society and in the companies could be influenced also by the situation in the country.

Response: Thank you for your kind advice. We have rewritten this section and clarified it as follows:

Second, because of industry constraints, most participants in this study were women. Moreover, all data were collected in state-owned enterprises in China, which limits the external validity of our findings. For instance, the prevailing managerial style in a society or country could be significantly influenced by the current situation in that country. Therefore, future research should expand the number and diversity of subjects, and explore whether the relationships identified here are observed in other industries and in a cross-cultural context. [p12, line413].

Reviewer 2 Report

Sample Notes:

- despite the size, we are dealing with only two enterprises

- (state) supervision does not have to be unambiguous - due to the employment policy in state-owned enterprises around the world - with supervision in private enterprises operating mainly in accordance with the principles of the free market in every aspect of activity.

- the structure of employees in terms of gender - is specific, as is the level of education, where 42.0% had an associate degree.

Scale - Lickert's scale is recognized in literature and science, but it has a major disadvantage - the grade '3', which in this type of study does not provide a positive or negative assessment. With a 7-point scale, this danger is clearly minimized.

These aspects are indicated in part by the Authors - they are aware of the limited possibilities of drawing conclusions based on the collected material.

Author Response

Thank you for taking time to provide this in-depth review of our paper. This valuable information has allowed us to improve the manuscript. Below, we clarify the changes that have been made.

1. Sample Notes:

Despite the size, we are dealing with only two enterprises (state) supervision does not have to be unambiguous - due to the employment policy in state-owned enterprises around the world - with supervision in private enterprises operating mainly in accordance with the principles of the free market in every aspect of activity.

the structure of employees in terms of gender - is specific, as is the level of education, where 42.0% had an associate degree.

Response: Thank you for your kind advice. We recognize the shortcomings of the sample and have described them in Limitations and Future Directions as shown below:

Second, because of industry constraints, most participants in this study were women. Moreover, all data were collected in state-owned enterprises in China, which limits the external validity of our findings. For instance, the prevailing managerial style in a society or country could be significantly influenced by the current situation in that country. Therefore, future research should expand the number and diversity of subjects, and explore whether the relationships identified here are observed in other industries and in a cross-cultural context [p12, line413].

2. Scale - Lickert's scale is recognized in literature and science, but it has a major disadvantage - the grade '3', which in this type of study does not provide a positive or negative assessment. With a 7-point scale, this danger is clearly minimized.

These aspects are indicated in part by the Authors - they are aware of the limited possibilities of drawing conclusions based on the collected material.

Response: Thank you for your kind advice. We have recognized the shortcomings of the method and have described them in Limitations and Future Directions as follows:

First, our data collection was carried out during a single time period; therefore, it might have been susceptible to common method bias (Podsakoff et al., 2003). Although the relevant empirical data support the theoretical hypotheses of this study, we recommend the use of multiple time points for data collection in future research in order to minimize the impact of common method bias effectively (Podsakoff et al., 2003) [p12, line408].

Reviewer 3 Report

Congratulations on your paper!

This work is very similar to the last one I reviewed from the same authors, so I propose the same changes.

In my opinion it is unnecessary to perform three different confirmatory factor analyzes. I think you should start directly from model 3, which incorporates all the constructs. Their results should be used to calculate other reliability indicators - in addition to Cronbach's alpha - such as composite reliability and the extracted variance. They should also be used to check the convergent and discriminant validity of the constructs. The latter is especially important given the high correlations between constructs reported in Table 2.

I think the paper would improve substantially with this change. I hope you

address the analysis of the convergent and discriminant validity of the scales.

on this occasion.

Author Response

Thank you for taking time to provide this in-depth review of our paper. This valuable information has allowed us to improve the manuscript. Below, we clarify the changes that have been made.

1. In my opinion it is unnecessary to perform three different confirmatory factor analyzes. I think you should start directly from model 3, which incorporates all the constructs. Their results should be used to calculate other reliability indicators - in addition to Cronbach's alpha - such as composite reliability and the extracted variance. They should also be used to check the convergent and discriminant validity of the constructs. The latter is especially important given the high correlations between constructs reported in Table 2. I think the paper would improve substantially with this change. I hope you address the analysis of the convergent and discriminant validity of the scales on this occasion.

Response: Thank you for your kind advice. Two measures were validated for the common method bias: Haman’s single factor method and confirmatory factor analysis. In the confirmatory factor analysis, we examined several alternative measurement models and compared them with the hypothesis model. The four-factor model fits our data better than other models, thus illustrating the discriminant validity between variables (Cangiano, Parker, & Yeo, 2019).

Cangiano, F., Parker, S. K., & Yeo, G. B. (2019). Does daily proactivity affect well‐being? The moderating role of punitive supervision. Journal of Organizational Behavior, 40(1), 59-72.

Reviewer 4 Report

There are 2 critical problems in the manuscript:

1)

The novelty of this manuscript is unclear. Although the introduction seemed to be well described, there are a lot of previous studies and very similar studies that the author has not mentioned.

e.g.)

https://doi.org/10.1080/1331677X.2018.1432374

http://hdl.handle.net/2097/15072

https://doi.org/10.1037/0021-9010.92.2.321

https://doi.org/10.1016/j.leaqua.2015.03.002

Authors need to do a more thorough review of existing literature to find out what is currently being revealed and what is unknown. And in the discussion, it is necessary to clarify what is new and outstanding in the dissertation.

2)

The study design is cross-sectional and causal relationships are unknown naturally. The author building the figure of the path (1) Abusive supervision -> (2) LMX -> (3) Job insecurity, however, whether these directions of the effects are true or not wasn’t examined and well assessed. A lack of LMX could cause an inadequate attitude of supervisors. A lack of Job security could damage the relationships between workers including supervisors and the others and also could affect the attitudes of supervisors. Thus, the methods and the results of the manuscript cannot make the conclusion that LMX has the mediating role and most of the descriptions in the discussion were overclaimed.

Minor issues:

3) The demographic variables and the results of the average of each score should be described.

4) The type of the company or the type of the job will affect the occupational variables. What kind of company did the author use? What kind of the job did they assigned? they How many participants were “supervisor”?

5) In china, in “state-owned enterprises”, is there a “job insecurity”? Can they get fired easily?

6) In the table 2, What kind of correlations analysis was used?

Author Response

Thank you for taking time to provide this in-depth review of our paper. This valuable information has allowed us to improve the manuscript. Below, we clarify the changes that have been made.

  1. The novelty of this manuscript is unclear. Although the introduction seemed to be well described, there are a lot of previous studies and very similar studies that the author has not mentioned.

Authors need to do a more thorough review of existing literature to find out what is currently being revealed and what is unknown. And in the discussion, it is necessary to clarify what is new and outstanding in the dissertation.

Response: Thank you for your kind advice. We have cited the following articles in the literature review:

At the same time, authentic leadership can reduce job insecurity by increasing psychological empowerment and psychological capital (Wang, Kan, Qin, Zhao, Sun, Mao, … Hu, 2020). [p2, line46]

In the Chinese context, job insecurity also brings many harms, such as emotional exhaustion and increased employee turnover intention (Wang, Hao, Zong, Siu, Xiao, Zhao, … Hu, 2020). [p2, line85]

Erdogan and Enders (2007) show that improvement of LMX can boost employees' job satisfaction and work performance, and that this relationship is moderated by supervisors' perceived organizational support. At the same time, Yildiz (2018) suggested that LMX reduces the turnover intention of employees by reducing mobbing behaviors. In addition, Xu et al. (2015) suggested that LMX can be used as a moderator variable to mitigate the negative effects of improper supervision. [p4, line152]

At the same time, we have also made some changes in the discussion as follows:

Although previous studies have explored the impact of LMX on turnover intention, the results of this study suggest that LMX is more likely to be a mediation mechanism that translates leaders’ negative behavior into employee’s job insecurity. By combining the social cognitive theory with the LMX theory (Liden & Maslyn, 1998; Ng & Lucianetti, 2016), we have clarified the internal mechanism of abusive supervision that affects employees’ job insecurities. [p11, line354]

  1. The study design is cross-sectional and causal relationships are unknown naturally. The author building the figure of the path (1) Abusive supervision -> (2) LMX -> (3) Job insecurity, however, whether these directions of the effects are true or not wasn’t examined and well assessed. A lack of LMX could cause an inadequate attitude of supervisors. A lack of Job security could damage the relationships between workers including supervisors and the others and also could affect the attitudes of supervisors. Thus, the methods and the results of the manuscript cannot make the conclusion that LMX has the mediating role and most of the descriptions in the discussion were overclaimed.

Response: Thank you for your kind advice. We are also aware of the shortcomings of cross-sectional data, which make our results likely to be affected by common method bias and fail to better illustrate the causal relationship between variables. We have mentioned this in the limitations section of the article. However, a series of tests have been carried out subsequently which demonstrate that the common method deviation in this study is not serious, meaning that the relationship between variables can be explained according to social cognitive theory and LMX theory. At the same time, the significance of the regression coefficient and mediating effect also support our hypothesis to some extent (Liu, Cheng, & Xin, 2018).

Liu, G., Cheng, Y., & Xin, Z. (2018). Three approaches to examine the mediating effect:a view of causal effect chain. Psychology: Techniques and Applications, 6(11), 665-676.

  1. The demographic variables and the results of the average of each score should be described.

Response: Thank you for your kind advice. We present the mean and standard deviation of each demographic variable in Table 2.

  1. The type of the company or the type of the job will affect the occupational variables. What kind of company did the author use? What kind of the job did they assigned? they How many participants were “supervisor”?

Response: Participants in this study were 936 employees from two state-owned enterprises located in China. The subjects were all front-line employees. [p5, line205].

  1. In china, in “state-owned enterprises”, is there a “job insecurity”? Can they get fired easily?

Response: Thank you for raising this interesting question. At the moment, the competition in state-owned enterprises is fierce. Younger employees can hardly adapt to the tense working conditions, and face the risk of being eliminated from the most recently filled position; older employees are also at risk of finding it hard to adapt to new technologies. Previous studies have shown that employees of state-owned enterprises in China have a certain degree of job insecurity (Wong, Wong, Ngo, & Lui, 2005).

Wong, Y.-T., Wong, C.-S., Ngo, H.-Y., & Lui, H.-K. (2005). Different responses to job insecurity of Chinese workers in joint ventures and state-owned enterprises. Human Relations, 58(11), 1391-1418.

  1. In the table 2, What kind of correlations analysis was used?

Response: Pearson correlation coefficient was used in table 2, the modified parts have been marked red [p8, line275].

Round 2

Reviewer 4 Report

Most of the concerns were improved, however, there still remain small problems of descriptions.
As the authors and the reviewer mentioned, this is a cross-sectional study at a single time period. The results provide only just a "correlation".
The word "predicts"(4.4, 4.5, and 7.conclusion) is not appropriate.

Author Response

  1. Most of the concerns were improved, however, there still remain small problems of descriptions.

As the authors and the reviewer mentioned, this is a cross-sectional study at a single time period. The results provide only just a "correlation".

The word "predicts"(4.4, 4.5, and 7.conclusion) is not appropriate.

Response: Thank you for your kind advice. We have rewritten this and clarified it as follows:

First, results of the regression analysis for job insecurity and abusive supervision indicated that abusive supervision had a significant and positive influence on job insecurity (β = 0.45, p < 0.001). [p8, line286].

The interaction between power distance and LMX had a significant impact on job insecurity (β = 0.13, p < 0.01), indicating that the moderated mediating effect was significant and hence that Hypothesis 3 was supported. [p9, line306].

The results of our study have shown that abusive supervision had a positive impact on job insecurity, and that LMX mediates the relationship between inappropriate supervision and job insecurity. [p12, line429].